

# Developing a multi-institutional nomogram for assessing lung cancer risk in patients with 5–30 mm pulmonary nodules: a retrospective analysis

Yongjie Jiang[1,*], Taibing Deng[2,*], Yuyan Huang[1], Bi Ren[1], Liping He[1], Min Pang[1] and Li Jiang[1]

[1] Department of Respiratory and Critical Care Medicine, The Affiliated Hospital of North Sichuan Medical College, Nanchong, Sichuan, China
[2] Department of Respiratory and Critical Care Medicine, Guang'an People's Hospital, Guang'an, Sichuan, China
[*] These authors contributed equally to this work.

Corresponding author
Li Jiang, lanqilily@163.com

## ABSTRACT

**Background**. The diagnosis of benign and malignant solitary pulmonary nodules based on personal experience has several limitations. Therefore, this study aims to establish a nomogram for the diagnosis of benign and malignant solitary pulmonary nodules using clinical information and computed tomography (CT) results.

**Methods**. Retrospectively, we collected clinical and CT characteristics of 1,160 patients with pulmonary nodules in Guang'an People's Hospital and the hospital affiliated with North Sichuan Medical College between 2019 and 2021. Among these patients, data from 773 patients with pulmonary nodules were used as the training set. We used the least absolute shrinkage and selection operator (LASSO) to optimize clinical and imaging features and performed a multivariate logistic regression to identify features with independent predictive ability to develop the nomogram model. The area under the receiver operating characteristic curve (AUC), $C$-index, decision curve analysis, and calibration plot were used to evaluate the performance of the nomogram model in terms of predictive ability, discrimination, calibration, and clinical utility. Finally, data from 387 patients with pulmonary nodules were utilized for validation.

**Results**. In the training set, the predictors for the nomogram were gender, density of the nodule, nodule diameter, lobulation, calcification, vacuole, vascular convergence, bronchiole, and pleural traction, selected through LASSO and logistic regression analysis. The resulting model had a $C$-index of 0.842 (95% CI [0.812–0.872]) and AUCs of 0.842 (95% CI [0.812–0.872]). In the validation set, the $C$-index was 0.856 (95% CI [0.811–0.901]), and the AUCs were 0.844 (95% CI [0.797–0.891]). Results from the calibration curve and clinical decision curve analyses indicate that the nomogram has a high fit and clinical benefit in both the training and validation sets.

**Conclusion**. The establishment of a nomogram for predicting the benign or malignant diagnosis of solitary pulmonary nodules by this study has shown good efficacy. Such a nomogram may help to guide the diagnosis, follow-up, and treatment of patients.

## INTRODUCTION

With the increasing popularity of CT screening and people's increased attention to physical examinations, the detection rate of pulmonary nodules has increased significantly (*Herder et al., 2005*; *Horeweg et al., 2014*). In the United States, incidental pulmonary nodules observed on chest CT account for 31% of cases (*Gould et al., 2015*). Lung cancer is the leading cause of cancer-related death worldwide and is one of the most common cancers (*Sung et al., 2021*). The five-year survival rate for lung cancer decreases with increasing stage, with a five-year survival rate as high as 92% for stage IA1 (*Goldstraw et al., 2016*). Therefore, the early diagnosis of malignant pulmonary nodules is particularly important for follow-up treatment and improving patient survival.

The nomogram is built based on a multifactorial regression analysis that integrates multiple predictive factors. It transforms the complex regression equation into a visual graph by converting the relationship between the variables by the calibrated line segment (*She et al., 2017*). Due to its readability and convenience, the nomogram has gained more attention and applications in clinical research and medical practice.

Foreign guidelines recommend the use of risk prediction models to assess the risk of lung cancer (*Gould et al., 2013*), but these models may not be applicable to all populations (*Bai et al., 2016*). Although some prediction models have been proposed in China in recent years (*Li et al., 2011*; *Liu et al., 2022*; *Zhou et al., 2022*), there is still no unified and clear prediction model that can be used clinically. Therefore, the purpose of this study is to establish and validate an appropriate nomogram for predicting the probability of malignant pulmonary nodules by combining clinical and CT imaging characteristics, with the aim of providing evidence-based support for the clinical management of pulmonary nodules.

## MATERIALS & METHODS

### Patients

The study retrospectively analyzed patients with pulmonary nodules at the Affiliated Hospital of North Sichuan Medical College and Guang'an People's Hospital between January 2019 and November 2021. Inclusion criteria required pulmonary nodules with a diameter of 5 mm to 30 mm, clear pathological diagnosis (benign nodules required surgical specimens, while malignant nodules required surgical or small biopsy specimens), and CT scan before pathological diagnosis. Exclusion criteria included completely calcified nodules, incomplete clinical information, and previous history of lung cancer. The study was approved by the Medical Ethics Committee of Affiliated Hospital of North Sichuan Medical College with file number 2022ER234-1. Informed consent was not obtained because this was a retrospective analysis where patient identity and privacy were protected. The process of patient selection is shown in Fig. 1.

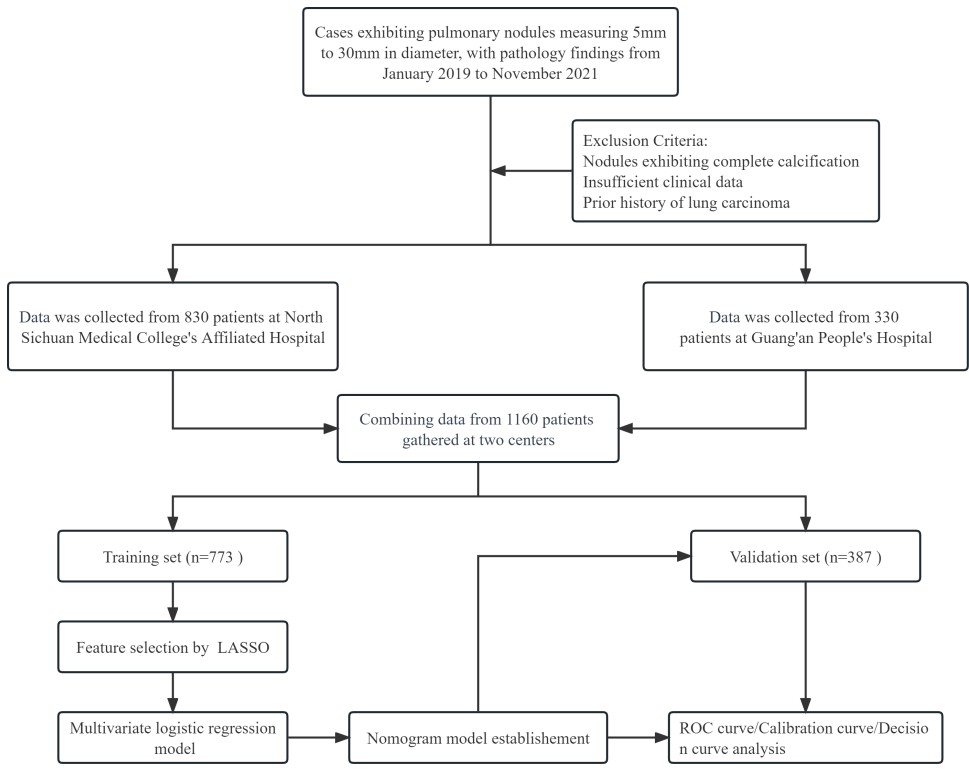

**Figure 1** The comprehensive flowchart, illustrating the process from patient selection to nomogram model establishment.

## Statistical analyses

Demographic and radiographic disease characteristics were presented as counts (%), and statistical analysis was performed using R software version 4.1.3.

The least absolute shrinkage and selection operator (LASSO) method is a type of linear regression that incorporates constraints to improve model accuracy. These constraints can prevent overfitting by shrinking the coefficients of less important features toward zero (*Nordhausen & Klaus, 2014*). Lasso regression is particularly useful in data sets with many features, some of which may not be relevant to the prediction task (*Friedman, Hastie & Tibshirani, 2010*; *Efron et al., 2008*). It is used to identify the most important features in a data set and reduce the complexity of the model (*Nordhausen & Klaus, 2014*). Variables selected by LASSO are included in multivariate logistic regression to identify independent predictors and develop the final model.

The calibration of the nomogram was evaluated using calibration curves, which assess the agreement between predicted probabilities and actual observed probabilities. The discriminative ability of the nomogram was assessed by Harrell's C-index and receiver operating characteristic (ROC) curves. Bootstrapping with 1,000 bootstrap resamples was used to validate the nomogram (*Kramer & Zimmerman, 2007*). Decision curve analysis (DCA) was performed to assess the clinical significance of the nomogram. DCA is a statistical method used to evaluate the clinical utility of diagnostic and prognostic strategies,
and it can indicate the net benefit of using the nomogram compared to alternative strategies (*Vickers & Elkin, 2006*).

## RESULTS

### Characteristics of patients

A total of 1,160 patients were included in this study, with 830 patients from the Affiliated Hospital of North Sichuan Medical College and 330 patients from Guang'an People's Hospital. Among the patients, the training set consisted of 773 individuals (372 males and 401 females), while the validation set comprised 387 individuals (172 males and 215 females). In the training set, 511 patients (66.11%) were diagnosed with malignant tumors, including 216 males (42.27%) and 295 females (57.73%). In the validation set, 288 patients (74.42%) were diagnosed with malignancies, including 119 males (41.32%) and 169 females (58.68%). A comprehensive overview of patient demographics and CT characteristics for both the training and validation sets is provided in Table 1.

### Feature selection

The study investigated the potential factors that influence the transformation of solitary pulmonary nodules into malignant tumors. Demographic data, including age, gender, smoking history, annual smoking volume, dust exposure history, family history of malignancy, and family history of lung cancer, and CT characteristics, including density of the nodule, nodule diameter, spiculation, edge, lobulation, shape, calcification, cavity, vacuole, vascular convergence, bronchiole, and pleural traction, were evaluated and included in the LASSO regression analysis. Thirteen features with non-zero coefficients were found after screening (Figs. 2A and 2B). These features included age, gender, family history of lung cancer, density of the nodule, nodule diameter, spiculation, lobulation, calcification, cavity, vacuole, vascular convergence, bronchiole, and pleural traction.

### Development of an individualized prediction model

In this study, logistic regression was used to identify predictors that could be used to differentiate between malignant and benign solitary pulmonary nodules. Thirteen features were initially analyzed, and the results showed that nine of these features could be used as independent predictors of malignancy: gender, density of the nodule, nodule diameter, lobulation, calcification, vacuole, vascular convergence, bronchiole, and pleural traction. Further analysis using multivariate logistic regression confirmed the importance of these factors, as summarized in Table 2. Finally, a nomogram was developed based on these nine predictors (Fig. 3).

### Nomogram model validation and clinical use

The calibration curves of the developed nomogram were evaluated in both the training (Fig. 4A) and validation (Fig. 4B) sets. The results showed good agreement between the predicted and actual probabilities, indicating a well-fitting model. In the training set, the $C$-index was 0.842 (95% CI [0.812–0.872]), and the AUC value (Fig. 4C) was 0.842 (95% CI [0.812–0.872]). The cut-off value of 0.504 yielded a maximum Youden index with a

**Table 1 Differences between demographic and CT imaging characteristics in the training and validation sets.**

| Demographic characteristics | Training set *n* (%) | | | Validation set *n* (%) | | |
|---|---|---|---|---|---|---|
| | Benignancy (*n* = 262) | Malignancy (*n* = 511) | Total (*n* = 773) | Benignancy (*n* = 99) | Malignancy (*n* = 288) | Total (*n* = 387) |
| Age (years) | | | | | | |
| <60 | 173 (66.03) | 308 (60.27) | 481 (62.23) | 67 (67.68) | 149 (51.74) | 216 (55.81) |
| >=60 | 89 (33.97) | 203 (39.73) | 292 (37.77) | 32 (32.32) | 139 (48.26) | 171 (44.19) |
| Gender | | | | | | |
| Male | 156 (59.54) | 216 (42.27) | 372 (48.12) | 53 (53.54) | 119 (41.32) | 172 (44.44) |
| Female | 106 (40.46) | 295 (57.73) | 401 (51.88) | 46 (46.46) | 169 (58.68) | 215 (55.56) |
| Smoking history | | | | | | |
| No | 173 (66.03) | 385 (75.34) | 558 (72.19) | 64 (64.65) | 221 (76.74) | 285 (73.64) |
| Yes | 89 (33.97) | 136 (26.61) | 225 (29.11) | 35 (35.35) | 67 (23.26) | 102 (26.36) |
| Annual smoking volume | | | | | | |
| <400 | 176 (67.18) | 385 (75.34) | 561 (72.57) | 64 (64.65) | 224 (77.78) | 288 (74.42) |
| >=400 | 86 (32.82) | 126 (24.66) | 212 (27.43) | 35 (35.35) | 64 (22.22) | 99 (25.58) |
| Dust exposure history | | | | | | |
| No | 255 (97.33) | 501 (98.04) | 756 (97.80) | 97 (97.98) | 285 (98.96) | 382 (98.71) |
| Yes | 7 (2.67) | 10 (1.96) | 17 (2.20) | 2 (2.02) | 3 (1.04) | 5 (1.29) |
| Family history of malignancy | | | | | | |
| No | 256 (97.71) | 491 (96.09) | 747 (96.64) | 93 (93.94) | 280 (97.22) | 373 (96.38) |
| Yes | 6 (2.29) | 20 (3.91) | 26 (3.36) | 6 (6.06) | 8 (2.78) | 14 (3.62) |
| Family history of lung cancer | | | | | | |
| No | 260 (99.24) | 494 (96.67) | 754 (97.54) | 98 (98.99) | 283 (98.26) | 381 (98.45) |
| Yes | 2 (0.76) | 17 (3.33) | 19 (2.46) | 1 (1.01) | 5 (1.74) | 6 (1.55) |
| **CT characteristics** | | | | | | |
| Density of the nodule | | | | | | |
| Pure ground-glass nodule | 29 (11.07) | 55 (10.76) | 84 (10.87) | 8 (8.08) | 35 (12.15) | 43 (11.11) |
| Mixed ground-glass nodule | 25 (9.54) | 156 (30.53) | 181 (23.42) | 11 (11.11) | 94 (32.64) | 105 (27.13) |
| Solid nodule | 208 (79.39) | 300 (58.71) | 508 (65.72) | 80 (80.81) | 159 (55.21) | 239 (61.76) |
| Nodule diameter(mm) | | | | | | |
| <10 | 87 (33.21) | 107 (20.94) | 194 (25.10) | 23 (23.23) | 55 (19.10) | 78 (20.16) |
| 10–20 | 135 (51.53) | 287 (56.16) | 422 (54.59) | 60 (60.61) | 176 (61.11) | 236 (60.98) |
| >20 | 40 (15.27) | 117 (22.90) | 157 (20.31) | 16 (16.16) | 57 (19.79) | 73 (18.86) |
| Spiculation | | | | | | |
| No | 195 (74.43) | 347 (67.91) | 542 (70.12) | 75 (75.76) | 189 (65.63) | 264 (68.22) |
| Yes | 67 (25.57) | 164 (32.09) | 231 (29.88) | 24 (24.24) | 99 (34.38) | 123 (31.78) |
| Edge | | | | | | |
| Rough | 219 (83.59) | 456 (89.24) | 675 (87.32) | 79 (79.80) | 263 (91.32) | 342 (88.37) |
| Smooth | 43 (16.41) | 55 (10.76) | 98 (12.68) | 20 (20.20) | 25 (8.68) | 45 (11.63) |
**Table 1** (*continued*)

| Demographic characteristics | Training set *n* (%) | | | Validation set *n* (%) | | |
|---|---|---|---|---|---|---|
| | Benignancy (*n* = 262) | Malignancy (*n* = 511) | Total (*n* = 773) | Benignancy (*n* = 99) | Malignancy (*n* = 288) | Total (*n* = 387) |
| Lobulation | | | | | | |
| No | 53 (20.23) | 69 (13.50) | 122 (15.78) | 18 (18.18) | 39 (13.54) | 57 (14.73) |
| Yes | 209 (79.77) | 442 (86.50) | 651 (84.22) | 81 (81.82) | 249 (86.46) | 330 (85.27) |
| Shape | | | | | | |
| Irregular | 237 (90.46) | 490 (95.89) | 727 (94.05) | 93 (93.94) | 275 (95.49) | 368 (95.09) |
| Regular | 25 (9.54) | 21 (4.11) | 46 (5.95) | 6 (6.06) | 13 (4.51) | 19 (4.91) |
| Calcification | | | | | | |
| No | 246 (93.89) | 509 (99.61) | 755 (97.67) | 92 (92.93) | 287 (99.65) | 379 (97.93) |
| Yes | 16 (6.11) | 2 (0.39) | 18 (2.33) | 7 (7.07) | 1 (0.35) | 8 (2.07) |
| Cavity(mm) | | | | | | |
| <5 | 258 (98.47) | 498 (98.4) | 756 (97.80) | 96 (96.97) | 276 (95.83) | 372 (96.12) |
| >=5 | 4 (1.53) | 13 (2.54) | 17 (2.20) | 3 (3.03) | 12 (4.17) | 15 (3.88) |
| Vacuole | | | | | | |
| No | 241 (91.98) | 441 (86.30) | 682 (88.23) | 97 (97.98) | 246 (85.42) | 343 (88.63) |
| Yes | 21 (8.02) | 70 (13.70) | 91 (11.77) | 2 (2.02) | 42 (14.58) | 44 (11.37) |
| Vascular convergence | | | | | | |
| No | 121 (46.18) | 69 (13.50) | 190 (24.58) | 42 (42.42) | 34 (11.81) | 76 (19.64) |
| Yes | 141 (53.82) | 442 (86.50) | 583 (75.42) | 57 (57.58) | 254 (88.19) | 311 (80.36) |
| Bronchiole | | | | | | |
| No | 240 (91.60) | 337 (65.95) | 577 (74.64) | 90 (90.91) | 180 (62.50) | 270 (69.77) |
| Yes | 22 (8.40) | 174 (34.05) | 196 (25.36) | 9 (9.09) | 108 (37.50) | 117 (30.23) |
| Pleural traction | | | | | | |
| No | 118 (45.04) | 177 (34.64) | 295 (38.16) | 43 (43.43) | 101 (35.07) | 144 (37.21) |
| Yes | 144 (54.96) | 334 (65.36) | 478 (61.84) | 56 (56.57) | 187 (64.93) | 243 (62.79) |

sensitivity of 0.611 and specificity of 0.918. In the validation set, the *C*-index was 0.856 (95% CI [0.811–0.901]), and the AUC value (Fig. 4D) was 0.844 (95% CI [0.797–0.891]). The highest Youden index of the model was observed at a cut-off value of 0.761, with a sensitivity of 0.828 and a specificity of 0.715. The non-significant difference in AUC ($P = 0.952$) between the validation and test sets indicated that the nomogram had excellent discrimination in internal validation. In addition, decision curve analysis (DCA) was performed on both the training (Fig. 4E) and validation (Fig. 4F) sets, indicating that using the nomogram to predict malignancy risk in lung nodules and intervening when the threshold probability is within a reasonable range may provide more benefit than intervening in all or none of the patients.

## DISCUSSION

In this retrospective study, we developed and validated a nomogram based on clinical and imaging features to differentiate early-stage lung cancer characterized by pulmonary nodules using logistic regression analysis. Gender, density of the nodule, nodule diameter,

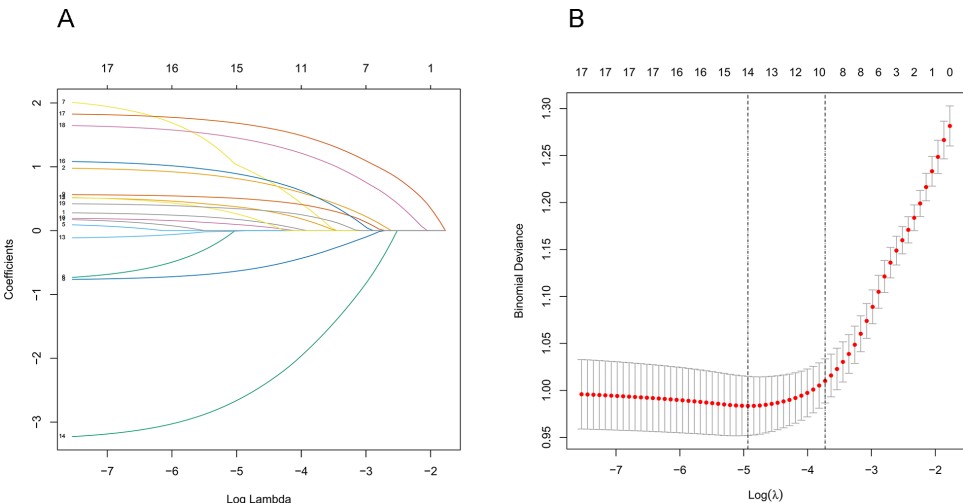

**Figure 2  Demographic and clinical feature selection using the LASSO binary logistic regression model.** (A) The model coefficient trend lines of the 19 features. Each line on the graph represents a variable, with the vertical axis showing the estimated coefficient of the variables and the ordinate showing the tuning parameter log (lambda) sequence. Different lambda values were used to identify different candidate variables, and the specific correlation coefficient for each measured variable was determined using coef (cvfit, s = lambda). The variables with non-zero coefficients were selected. (B) Depiction of the process of selecting optimal parameters by LASSO regression. The abscissa represents the logarithm of the parameter λ, while the ordinate represents the model errors. The numbers at the top of the figures indicate the number of candidate variables for the corresponding lambda value in LASSO regression. Lambda weighting parameters of λ = 0.0087 were considered optimal, and thirteen non-zero radiomics features were ultimately selected to construct the model. Abbreviations: LASSO, least absolute shrinkage and selection operator.

lobulation, calcification, vacuole, vascular convergence, bronchiole, and pleural traction were identified as the most valuable predictors for identifying malignant lung nodules. The nomogram showed high accuracy and robustness in both the training and validation sets. In the training set, the AUC of the nomogram was 0.842, indicating an accuracy rate of approximately 84% in predicting the probability of malignancy in clinical pulmonary nodules. Robustness was confirmed in the validation set with an AUC of 0.844. The calibration curve showed good agreement between the predicted and actual probabilities in both the training and validation sets, suggesting that the model has a high degree of calibration. Decision curve analysis revealed our nomogram would be beneficial in guiding decisions about scenarios where all patients receive no intervention or all patients receive intervention when the threshold probability is within a reasonable range.

Physicians rely primarily on visual assessment of two-dimensional CT images to evaluate pulmonary nodules, which inevitably leads to individual errors (*He et al., 2021*). Consequently, differences in personal experience among different physicians may lead to controversies in clinical practice. Therefore, it is crucial to establish quantitative relationships between clinical features, radiographic findings, and pathologies to support accurate clinical decision making. Our nomogram accurately quantifies clinical and imaging
**Table 2  Multivariate analysis of variables in the predictive model.**

| Intercept and variable | Prediction model | | |
|---|---|---|---|
| | β | Odds ratio (95% CI) | P-value |
| (Intercept) | −2.570 | 0.077 (0.035–0.166) | <0.001 |
| Age: >=60 (vs <60) | 0.250 | 1.284 (0.858–1.930) | 0.226 |
| Gender: Woman (vs Man) | 0.912 | 2.489 (1.688–3.700) | <0.001 |
| Family history of lung cancer: Yes (vs No) | 1.528 | 4.609 (1.156–31.252) | 0.057 |
| Density of the nodule (vs pure ground-glass) | | | |
| Mixed ground-glass | 1.185 | 3.269 (1.573–6.883) | 0.002 |
| Solid | −0.883 | 0.414 (0.220–0.761) | 0.005 |
| Nodule diameter (vs <10 mm) | | | |
| 10 mm–20 mm | 0.494 | 1.639 (1.029–2.617) | 0.038 |
| >20 mm | 1.074 | 2.926 (1.580–5.511) | <0.001 |
| Spiculation (vs No) | 0.230 | 1.258 (0.818–1.944) | 0.297 |
| Lobulation (vs No) | 0.563 | 1.756 (1.013–3.056) | 0.045 |
| Calcification (vs No) | −3.904 | 0.020 (0.002–0.114) | <0.001 |
| Cavity (vs No) | 0.619 | 1.857 (0.573–7.544) | 0.336 |
| Vacuole (vs No) | 1.067 | 2.908 (1.571–5.605) | <0.001 |
| Vascular convergence (vs No) | 1.914 | 6.782 (4.409–10.608) | <0.001 |
| Bronchiole (vs No) | 1.690 | 5.419 (3.229–9.486) | <0.001 |
| Pleural_traction (vs No) | 0.506 | 1.658 (1.079–2.555) | 0.021 |

**Notes.**
β is the regression coefficient.

features and can effectively guide clinicians in evaluating and making treatment decisions for pulmonary nodules.

Numerous studies have demonstrated that lobulation is a reliable factor for differentiating benign from malignant pulmonary nodules (*Qi et al., 2020*; *Chu et al., 2020*; *Liu et al., 2020*). In our study, we also found that lobulation (OR =1.756, 95% CI [1.013–3.056]) was an independent risk factor for such differentiation. Furthermore, our research indicated that gender (OR =2.489, 95% CI [1.688–3.700]) is another predictor of benign and malignant pulmonary nodules, with females having a higher likelihood of developing malignant nodules. This is consistent with recent studies suggesting that lung adenocarcinoma, a more common type of lung cancer than squamous cell carcinoma (*Austin et al., 2013*), typically affects women more often than men due to differences in disease susceptibility and risk factors (*Bai et al., 2016*; *Kinoshita et al., 2017*). Our study also found that solid nodules (OR =0.414, 95% CI [0.220–0.761]) were a protective factor, whereas mixed ground-glass nodules (OR =3.269, 95% CI [1.573–6.883]) were a risk factor. This indicates that ground-glass nodules are more likely to be malignant than solid nodules. Research by *Henschke et al. (2002)* supports this finding, showing that mixed ground-glass nodules had a malignancy rate of 63%, compared to 18% for pure ground-glass nodules and 7% for solid nodules. In addition, vascular convergence sign, an indicator of malignant tumor growth and metastasis (*Nishida et al., 2006*; *Raghu et al., 2019*), was also identified

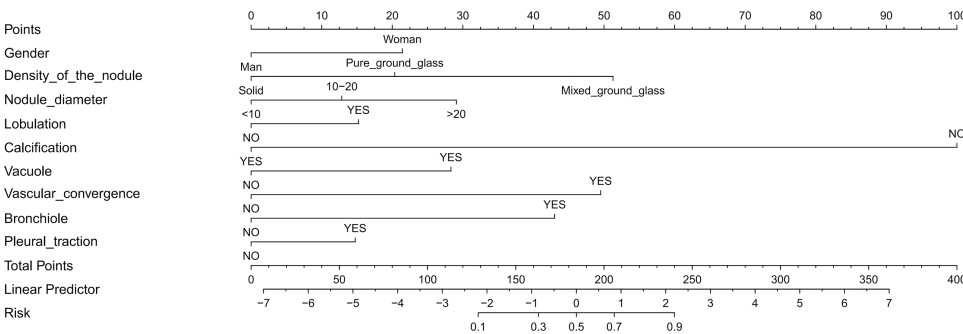

**Figure 3** **Nomogram to predict the probability of pulmonary nodules being malignant in patients.**
Each variable is represented by a marked line segment on a scale representing its possible range of values,
with the length of the line reflecting its contribution to the outcome event. Each variable was assigned a
score on the point scale axis, and a total score was calculated by summing these individual scores. By pro-
jecting the score to the lower end of the total point scale, we could estimate the probability that the pa-
tients' pulmonary nodules were malignant.

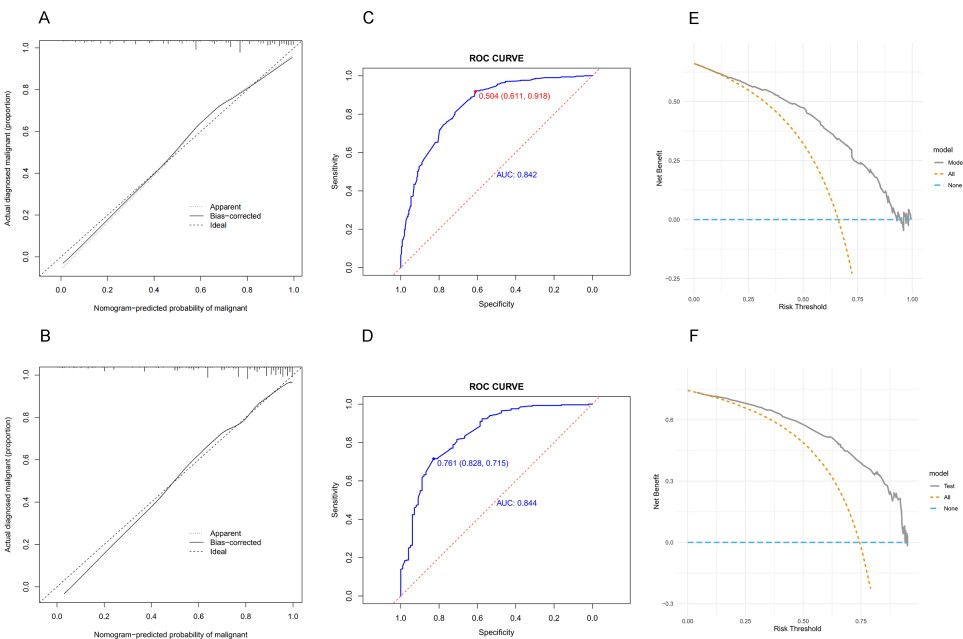

**Figure 4** **The predictive value and clinical use of the nomogram model for predicting the probability
of pulmonary nodules being malignant in the training and validation sets.** (A) Calibration of the train-
ing set. (B) Calibration of the validation set. (C) Discrimination of the training set. (D) Discrimination of
the validation set. (E)Decision curve analysis of the training set. (F) Decision curve analysis of the valida-
tion set.

as an independent risk factor for pulmonary nodules in our study, consistent with previous
research (*Huang et al., 2017*; *Wang et al., 2018*).

As nodular diameter increases, so does the probability of malignancy (*MacMahon et al.,
2017*). Nodules with a maximum diameter of less than five mm rarely have a malignancy

rate above 0.4%, while nodules between 5 and 10 mm have a 1.3% probability of being cancerous. Nodules larger than 10 mm have a much higher malignancy rate of 15.2% (*Horeweg et al., 2014*). In our study, we found that as the diameter of nodules increased between 10 mm and 20 mm, the risk of malignancy increased by approximately 0.6 times for each 1 mm increase in diameter. When nodules exceeded 20 mm in diameter, the risk of malignancy increased by approximately 1.9 times per 1 mm increase. Calcification was identified as a protective factor in our model. This is because calcification is typically associated with the healing of old lesions and represents stable, benign lesions (*Gorospe et al., 2020*). Features such as diffuse, layered, and central calcifications are highly suggestive of benign lesions, such as inflammatory pseudotumors, whereas only a small percentage of malignant nodules exhibit eccentric or punctate calcifications (*Zhou et al., 2021*). Our study also included several characteristic signs of malignant pulmonary nodules. These include vacuole (OR =2.908, 95% CI [1.571–5.605]), bronchiole (OR =5.419, 95% CI [3.229–9.486]), and pleural traction (OR =1.658, 95% CI [1.079–2.555]), which are consistent with the findings of previous studies (*Xia et al., 2021*; *Zhao et al., 2021*; *Hou et al., 2021*).

Our study has several advantages. First, we developed a nomogram to build prediction models, which facilitates the quantification of risks and provides more intuitive results. Second, all nodules included in our study had clear pathologic diagnoses, ensuring that our results were based on accurate data. Third, by screening data from two research centers, our study was able to obtain a larger volume of data and broader coverage, resulting in a prediction model that is more universally applicable. Fourth, our model was developed based on CT and clinical features, which makes the scoring factors for pulmonary nodules simple and easy to obtain. In comparison, other models that incorporate radiomics, ctDNA, and serum tumor markers may be less accessible to clinicians. As such, our model may be easier to disseminate, particularly to outpatient physicians who need a quick and accurate assessment of benign and malignant pulmonary nodules to make informed decisions about diagnosis and treatment.

This study has several limitations. First, our study was retrospective, and further verification of its performance will be required in future prospective studies. Second, although the internal validation of the model showed excellent stability and calibration, the external validation of the model using separate data still needs to be performed in the future. Third, in order to make the model more concise, accessible, and user-friendly, we did not include radiomics, ctDNA, and serum tumor markers in our study. However, the inclusion of these factors may help to improve the accuracy of the model in the future.

## CONCLUSIONS

A nomogram based on clinical and CT characteristics was developed and validated in our study to quantify the probability of pulmonary nodule presenting as early lung cancer. This nomogram has potential value in the clinical management of pulmonary nodules.

## ACKNOWLEDGEMENTS

We are grateful to Haiying Gong and Fang He for helpful discussions and thoughts.

### Funding

This work was supported by the Guang'an City Pulmonary Nodule/Lung Cancer Whole Process Management Study (2020SYF03). The funders had no role in study design, data collection and analysis, decision to publish, or preparation of the manuscript.

### Grant Disclosures

The following grant information was disclosed by the authors:
The Guang'an City Pulmonary Nodule/Lung Cancer Whole Process Management Study: 2020SYF03.

### Competing Interests

The authors declare there are no competing interests.

### Author Contributions

- Yongjie Jiang conceived and designed the experiments, performed the experiments, analyzed the data, prepared figures and/or tables, authored or reviewed drafts of the article, and approved the final draft.
- Taibing Deng conceived and designed the experiments, authored or reviewed drafts of the article, and approved the final draft.
- Yuyan Huang conceived and designed the experiments, performed the experiments, analyzed the data, prepared figures and/or tables, authored or reviewed drafts of the article, and approved the final draft.
- Bi Ren performed the experiments, authored or reviewed drafts of the article, and approved the final draft.
- Liping He performed the experiments, authored or reviewed drafts of the article, and approved the final draft.
- Min Pang performed the experiments, authored or reviewed drafts of the article, and approved the final draft.
- Li Jiang conceived and designed the experiments, authored or reviewed drafts of the article, and approved the final draft.

### Human Ethics

The following information was supplied relating to ethical approvals (i.e., approving body and any reference numbers):

The study was approved by the Medical Ethics Committee of Affiliated Hospital of North Sichuan Medical College with file number 2022ER234-1.

## Data Availability

The data is available at Figshare: Jiang, Yongjie (2023). DATA. figshare. Dataset. Available at https://doi.org/10.6084/m9.figshare.22581355.v3.

## Supplemental Information

Supplemental information for this article can be found online at http://dx.doi.org/10.7717/peerj.16539#supplemental-information.

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
