# Peer review of "Developing a multi-institutional nomogram for assessing lung cancer risk in patients with 5–30 mm pulmonary nodules: a retrospective analysis"

_PeerJ, doi:10.7717/peerj.16539_

## Round 0.1 · original submission · Major Revisions

Please make revisions to address all major concerns of reviewers, especially on the sample size and clarifications on the findings.

·

Basic reporting

OK

Experimental design

Please add a screening flowchart.

Validity of the findings

OK

Additional comments

Thank you for you innovate study attempting for identifying lung cancer risk in patients with pulmonary nodule. I have a suggestion that the author divide the validation cohort into two parts, including an internal validation cohort and an external validation cohort.

Reviewer 2 ·

Basic reporting

The manuscript is generally clear and easy to understand. The English language could be further improved. The article structure is ok. Raw data shared.

Experimental design

1. Please rationale the sample size.
2. How do you determine the Clinical and CT imaging features to be included?

Validity of the findings

1. Has there been any previous publication about similar nomogram model? If yes, please state the superiority or differences to other studies.
2. As stated in the limitation, prospective study and external validation should be conducted to further validate the results.

Reviewer 3 ·

Basic reporting

In this manuscript, the authors retrospectively analyzed the clinical and CT data of 1160 patients with pulmonary nodules (5-30mm) in two hospitals. Patients were randomly divided into the training cohort and the validation cohort. Using LASSO regression and several other methods, the authors found that the sex, density of the nodules, nodule diameter, lobulation, calcification, vacuole, vascular convergence, bronchiole, and pleural traction were predictive variables. Then the authors established a nomogram to predict the risk of lung cancer using those predictive variables.

Those predictive variables have been frequently reported in predicting the risk of lung cancer, impairing the novelty of this research. Meanwhile, deep learning has greatly changed the diagnosis of lung cancer, with high accuracy and specificity.

Experimental design

Patients included in this research had to receive surgical specimens or small biopsy specimens, which may cause great bias, since most patients with less lung cancer risk would not receive surgery or biopsy.

Validity of the findings

no comment.

Additional comments

The title should be more precise, since only patients with a diameter of 5-30mm pulmonary nodules were included. The number of patients from each institution should be provided.

---

## Round 0.2 · Minor Revisions

A reviewer still has a concern regarding a discussion point. Please address the reviewer's question in the revision.

Reviewer 2 ·

Basic reporting

The authors have addressed the comments satisfactorily.

Experimental design

The authors have addressed the comments satisfactorily.

Validity of the findings

The authors have addressed the comments satisfactorily.

Reviewer 3 ·

Basic reporting

Why do patients with solid nodules have lower cancer risk than those with pure GGO? It seems irrational. Please discuss it.

Are nodules with spiculation or lobulation classified into irregular nodules?

Experimental design

no comment

Validity of the findings

no comment

Additional comments

no comment

---

## Round 0.3 · accepted · Accept

The revised version addresses reviewers questions well and is ready for publication.